# Effects of Molarity and Storage Time of MWCNTs on the Properties of Cement Paste

**DOI:** 10.3390/ma15249035

**Published:** 2022-12-17

**Authors:** Echeverry-Cardona Laura, Cabanzo Rafael, Quintero-Orozco Jorge, Castillo-Cuero Harvi Alirio, Rodríguez-Restrepo Laura Victoria, Restrepo-Parra Elisabeth

**Affiliations:** 1Laboratorio de Física del Plasma, Universidad Nacional de Colombia, Sede Manizales, Manizales 170001, Colombia; 2Laboratorio de Espectroscopía Atómica y Molecular (LEAM), Centro de Materiales y Nanociencias (CMN), Parque Tecnológico Guatiguará, Universidad Industrial de Santander, Bucaramanga 681012, Colombia; 3Ciencia de Materiales Biológicos y Semiconductores (CIMBIOS), Centro de Materiales y Nanociencias (CMN), Parque Tecnológico Guatiguará, Universidad Industrial de Santander, Bucarmanga 681012, Colombia; 4Centro de Nanociencias y Nanotecnología, Universidad Nacional Autónoma de México, Km 107 Carretera Tijuana Ensenada, Ensenada 22860, Mexico

**Keywords:** MWCNT, dispersion, sonication energy, stability, cement paste

## Abstract

Nowadays, nanomaterials in cement pastes are among the most important topics in the cement industry because they can be used for several applications. For this reason, this work presents a study about the influence of changing the molarity of dispersed multiple wall carbon nanotubes (MWCNTs) and varying the number of storage days on the mechanical properties of the cement paste. To achieve this objective, dispersions of 0.35% MWCNTs, varying the molarity of the surfactant as 10 mM, 20 mM, 40 mM, 60 mM, 80 mM, and 100 mM, were performed. The mixture of materials was developed using the sonication process; furthermore, materials were analyzed using UV-Vis, Z-potential, and Raman spectroscopy techniques. Materials with a molarity of 10 mM exhibited the best results, allowing them to also be stored for four weeks. Regarding the mechanical properties, an increase in the elastic modulus was observed when MWCNTs were included in the cement paste for all storage times. The elastic modulus and the maximum stress increased as the storage time increased.

## 1. Introduction

Carbon nanotubes (CNTs) have been used in a variety of applications due to their versatility. Some of their applications are as additives in polymers, catalysts, autoelectron emission for cathodic rays in illumination components, absorption and filters of electromagnetic waves, energy conversion, anodes of lithium batteries, hydrogen storage, and sensors, among others [1]. CNTs are a carbon allotrope phase that possesses intermediate properties between graphite and fullerenes [1,2]. These materials are composed of sp^2^ hybridization carbon bonds and can be produced as structures with a simple wall or multiple walls separated by around 0.35 nm [3]. In this sense, multiple wall carbon nanotubes (MWCNTs) have gained more attention due to their high performance and low cost of production per unit. In addition, their thermodynamic stability and capacity to sustain and improve the electrical properties make them excellent candidates for applications that require these special properties [3].

For instance, the cement industry has used MWCNTs as an additive in cement matrices to improve the electrical and mechanical properties [4]. MWCNTs are added in proportions of 0.2% of the cement weight to enhance the flexural strength; this is an important aspect that may have to be taken into account, because when the mixture is performed using traditional mechanical methods, if they are included in higher ratios, the MWCNT dispersion can present strong drawbacks such as exhibiting possible agglomerations and clusters [5]. To avoid these problems, the ultrasound technique is mostly used because it fragments the MWCNT agglomerations [6,7,8,9]. The dispersion is perhaps the most critical factor that influences the mechanical properties of cement pastes [10]. Some experiments have confirmed that MWCNTs can be effectively dispersed in water using ultrasound energy and commercial surfactants [11,12,13]. In the literature, it is possible to observe works focused on applying surfactants to sustain the dispersion [14]. For example, Mendoza [15] studied several dispersion concentrations of surfactants such as sodium lauryl sulfate, cetylpyridinium chloride, and Triton X-100, finding that the reinforcing effect of MWCNTs is masked by the negative effect of the surfactants.

The main limitations of using MWCNTs are (i) achieving a total dispersion in water and (ii) reaching the stability for a long storage time [16]. Because the van der Waals forces are responsible for these phenomena, the size of the agglomerates can reach the micrometer scale [17,18,19]. Furthermore, these agglomerations can cause a stress concentration because they behave as weak spots in the cement paste, reducing the fluidity of the material because they absorb the free water [20]. Because of this, different investigations and several approaches have been proposed for their dispersion [21,22,23]. Physical methods such as sonication and adsorption of the surfactant are the most currently used. Regarding the use of surfactants, they are considered a surface-active agent that has an amphipathic structure, containing a lyophobic (solvent repulsive) and a lyophilic (solvent attractive) group. It has been found that using low surfactant concentrations allows the molecules to be absorbed on the surface or interface, decreasing the interfacial tension and improving the dispersion [24]. This is due to the variation in the dielectric constant of the water depending on the surfactant type used; for example, if it is an ionic surfactant, the particles are stabilized via a repulsive electrostatic force, whereas if it is a nonionic surfactant, an interparticle repulsion via steric-hydration forces is produced [25].

Researchers have evaluated nonionic surfactants for dispersing MWCNTs, graphene, and graphene oxide, among others [26,27,28,29,30,31,32]; nevertheless, these studies have focused on using different surfactants without considering changes in the concentration of the surfactant. For example, Blanch et al. [25] reported than the increase in the surfactant concentrations above a certain value led to the flocculation of the CNTs, possibly due to the attractive depletion interactions. This effect generated a poor dispersion. The surfactant concentration influences the dispersion, generating an encapsulation of MWCNTs in cylindrical micelles, adsorption of hemimicelles, or random adsorption; then, each surfactant must be working below the critical micelle concentration. Although it has been demonstrated that optimal concentrations exist for nanotube dispersion [33], few studies have presented a more in-depth analysis of the influence of the surfactant concentration on nanotube dispersion.

In this work, we have produced dispersions of MWCNTs with Triton TX-100 surfactant for introduction into the cement matrix. A detailed analysis of the Triton X-100 concentration for values of 10, 20, 40, 60, 80, and 100 mM was carried out to find the optimal concentration. Firstly, UV-Vis, Z-potential, and Raman spectroscopy analyses were performed to determine the dispersion at different storage times (1, 2, and 4 weeks), and secondly, three types of cylinders of cement paste were produced: (i) without MWCNTs, (ii) with MWCNTs + TX-100 (one week of storage), and (iii) with MWCNTs + TX-100 (four weeks of storage).

## 2. Materials and Methods

### 2.1. Materials

For the samples built, a mixture of water type 1, MWCNTs, and Triton TX-100 was carried out.

Water type I was used because it is required to avoid elements that alter the electrical properties of the surfactant and MWCNTs.

Triton TX-100 was used as a surfactant because it exhibits a nonionic character and contributes negative charges; thus, it does not affect the electrostatic repulsion or attraction of the nanotubes. 

Finally, industrial grade MWCNTs NC7000 produced by Belgium Nanocyl SA were used.

### 2.2. Dispersion Procedure

In order to identify the influence of the percentage of surfactant on the behavior of the mixture, the Triton TX-100 surfactant was used with molarities of 10 mM, 20 mM, 40 mM, 60 mM, 80 mM, and 100 mM. The percentage of the MWCNTs was chosen as 0.35%, according to the literature [15]. The steps performed during the materials production were:-The TX-100 at different molarities (10 mM, 20 mM, 40 mM, 60 mM, 80 mM, and 100 mM) and water type 1 were mixed for 5 min using a magnetic stirrer at room temperature;-After that, MWCNTs were added;-The mixture was placed in the ultrasonic cube with a power of 500 W and 40% amplitude, applying an energy of 390 J/g;-Sonication was performed with 20-s on/off cycles;-The room temperature was kept constant using a cold bath that consists of immersing the beaker containing the mixture in a larger beaker containing a mixture of ice and water. The temperature was constantly measured and maintained at room temperature;-Materials were stored for 1, 2, 4, 10, and 13 weeks;-After that, the materials were characterized in order to determine the stability of the samples;-The test cylinders were made from the mixtures stored for 1 and 4 weeks because for 2 weeks, the results were very similar to those of 1 week, and for 10 and 13 weeks, the material had already become unstable, according to the UV-Vis spectroscopy analysis.

The sonication time ($t_{son}$) was calculated with the relationship obtained by Mendoza-Reales [11]. According to this report, the time for dispersing 155 g is 60 min, with an energy ratio of 390 J/g. With these values, the calculated energy is 390 J/g × 155 g = 60,450 J. From this reference, the proposed time of sonication is:(1)$$t_{son}=\frac{\left(\left(m_{water}+m_{dis}+m_{MWCNT}\right)*E_{dis}\right)*60}{60,450\;J}$$
where *m_water_*, *m_dis_*, and *m_MWCNT_* are the masses of the water, dispersant, and MWCNTs, respectively, and *E_dis_* is the dispersion energy. Table 1 shows the values of the total mass, dispersion energy, and sonication time for the experiments. A diagram of the experimental setup is shown in Figure 1.

### 2.3. Materials Characterization

A UV-Vis UV2600 (Shimadzu—Chicago, IL, USA) with a 200 to 850 nm spectral range was used to obtain the UV-Vis spectra. Z sizer nano Ze3690 de Malvern was used to obtain the Z-potentials, with water as the solvent and 1.33 refraction indices. For both UV-Vis and Z sizer characterizations, the samples had to be diluted in a ratio of 1 to 100 in type 1 water. Moreover, the measurements were obtained for the six samples by varying the molarity of the surfactant. Furthermore, the measurements on all samples were carried out by varying the weeks of storage (1, 2, 4, 10, and 13 weeks). Each measurement was performed five times for statistical purposes, and the average value and standard deviation were determined.

A Raman Confocal LabRam HR Evolution, Horiba Scientific (YOBIN IVON), was used to obtain the Raman spectra with the following conditions: 532 nm laser, optical microscopy with 10X magnification, and a 1250–1690 cm^−1^ spectral range. Raman spectra were taken by varying the molarity of the TXT-surfactant. Data were acquired for samples stored for one week. For one week, the mixtures exhibited the highest stability; in addition, in the case of using the materials in a particular application, costs must be reduced—for example, those related to storage. This means that the mixture with greater stability was selected, which implies less economic and time costs.

### 2.4. Construction of the Test Cylinders

Cylinders of the cement paste with a H_2_O/cement ratio of 0.4 were built according to the following equation:(2)$$\frac{\mathrm{MWCNTs}/\mathrm{TXT}-100/H_{2}O}{\mathrm{Cement}}=0.4$$

The cylinders were built with a 1-inch diameter and a 2-inch length (C109/C109M ASTM norm) [34]. The samples were made with the NTC 550 norm. Firstly, the cement was mixed with the MWCNTs/TX-100/H_2_O solution; secondly, the mixture was introduced into the cylinders through three equal layers using the compaction method. This was carried out using 50 beats for each layer to decrease the porosity. Finally, the cylinders were brought to room temperature and, after 24 h, were introduced to a calcium oxide-cured process (see Figure 2), according to ASTM C192 norm [35]. Figure 3a shows the specimens during the drying process, while Figure 3b presents a photograph of the specimens in the curing and storage processes.

### 2.5. Properties of the Cylinders

Finally, a Humbolt HM 5030 Master Loader (Manizales, Colombia) with a 50 kN capacity load cell was used in the test of the specimens. The established parameters were a speed of 0.25 mm/min, taking the strain data every 0.010 mm, and measuring the load in kN for each strain reading [36,37]. The elasticity modulus and maximum strength were obtained from the stress—strain curve. The parameters used were 0.25 mm/min velocity and data taken each 0.010 mm. A Carl Zeiss EVO MA 10 scanning electron microscope (Oxford model Xact) equipped with a silicon detector of 10 mm was used for morphological examination. Images were taken with a resolution of 5 nm.

## 3. Results

The first result analyzed was the degree of dispersion of the MWCNTs into the cement using UV-Vis spectroscopy. This analysis was carried out on the samples of MWCNTs mixed with type 1 water and with the TXT-100 dispersant, varying the molarity and storage time. The degree of dispersion of the nanotubes within the TXT-100 dispersant is directly related to the presence of peaks in the spectra, which are an indication of the generation of certain bonds, as will be explained later.

The maximum absorbance in the UV-Vis spectra was identified at 300 nm. It is well known that the agglomerated CNTs absorb in the ultraviolet region at around 300 nm; meanwhile, the individual CNT is active in the Vis region. Hence, it is possible to establish a relationship between the absorbance intensity and the degree of dispersion [36,37]. Moreover, the behavior of each sample was evaluated by varying the number of weeks of storage and determining the stability as a function of time. Figure 4 shows the intensity of the absorbance peak at 300 nm as a function of the molarity and the weeks of storage using UV-Vis. According to this figure, when the molarity is increased, the intensity of the maximum absorbance (at 300 nm) increases. This behavior is due to the presence of a great quantity of benzene rings and alkyne chains that causes many interactions between the surfactant and the MWCNTs (π-π stacking and van der Waals forces). This effect would entail a higher π plasmon resonance [38,39].

The presence of the peak in the UV-Vis spectra is an indication that there is good dispersion and integration between the MWCNTs and the TXT-100 dispersant. By way of their hydrophobic group, the surfactants get adsorbed onto the exterior surface of the MWCNT via noncovalent attraction forces [40], including hydrophobic interaction, hydrogen bonding, π–π stacking, and electrostatic interaction [41], which improve the dispersion of CNTs through steric or/and electrostatic repulsion [42]. It should be noted that the solutions (H_2_O + TX-100 + MWNTC), varying the TXT-100 molarity, exhibit a good dispersion (stability) for 1, 2, and 4 weeks, showing a high intensity of the absorbance peak at 300 nm. Nevertheless, at the 10th and 13th weeks, the intensity of the absorbance peak abruptly decreases to zero, indicating that the MWCNTs were agglomerated. This is a promising result because as far as we know, there have been no studies about the time for which the solution (H_2_O + TX-100 + MWNT) remains active. The fact that the absorbance intensity decreases as a function of the weeks of storage indicates that the MWCNTs remained dispersed for a few weeks of storage. For many weeks of storage, the nanotubes tend to agglomerate. MWCNTs produce small clusters/agglomerates due to their high affinity. As the storage time increases, the nanotubes tend to agglomerate, taking into account that just as other nanostructures, they have a large number of free bonds on the surface that are highly reactive. This high reactivity generates a strong attraction between them, causing them to get closer until they agglomerate [18].

On the other hand, the Z-potential spectra were obtained for 1, 2, 4, 10, and 13 weeks for samples with varying TXT-100 molarity. The Z-potential is related to the surface of hydrodynamic shear. When the MWCNTs are immersed in the surfactant, the surfactant layer surrounding the nanotubes can be divided into two parts: an inner region (Stern layer) where the ions are strongly bonded and an outer (diffuse) region where they are less bonded. There exists a notional boundary within the diffuse region where the ions and the nanotubes form a stable interaction. When a nanotube moves, for instance, due to the gravitational force, the boundary is shifted due to the ion movement. Those over-the-limit ions remain with the TXT-100 bulk dispersant. The potential formed at this boundary region is named the Z-potential. A schematic representation for the nanotubes is presented in Figure 5 [43].

The value of the Z-potential gives an indication of the potential stability of the colloidal system. If the nanomaterials in suspension exhibit a large negative or positive zeta potential, they tend to repel each other, avoiding agglomeration. Nevertheless, if nanomaterials present a low zeta potential value, no forces exist to prevent the nanomaterial agglomeration and flocculation. Nanomaterials with Z-potentials more positive than +30 mV or more negative than −30 mV are stable; this depends on the type of dispersant. Considering these aspects, the Z-potential analyses were carried out.

Figure 6 shows an increasing tendency of the Z-potential when the molarity and storage period are increased. In the case of the increase in molarity, the amount of moles present in solution of course increases. This makes that the tense-active micelles produce a decrease in the structural damage and the electrostatic charges present in the MWCNTs surface [38]. It is known that when a minor electrostatic charge is present in the MWCNT surface, the repulsive force and electrostatic attractions also decrease; furthermore, as the molarity of the surfactant is increased, a great quantity of mass is found around the MWCNTs, avoiding their agglomeration. On the other hand, as the period of storage is increased, the Z-potential decreases. It can be explained through the minimum energy principle: a stable system can experience instability when it is subjected to external energy; nevertheless, when this energy is suppressed, the system comes back to its initial state. This instability is generated by electrostatic charges present in the MWCNT surface. This electrostatic charge is produced by the rupture of energetically weak bonds [44,45]. Furthermore, it is well known that colloids tend to precipitate because of the gravitational force; then, as the number of weeks increases, the MWCNTs tend to agglomerate due to the precipitation, and the Z-potential decreases drastically. This means that the samples with better conditions for building the test cylinder in order to determine the mechanical properties are those cured for 1, 2, and 4 weeks, taking into account that the sample cured for one week exhibited the greater Z-potential values. On the other hand, for the case of four weeks, an intermediate behavior was observed. Then, these two samples were chosen for the next stage of the experiment; that is, the mechanical properties evaluation.

Before the mechanical properties evaluation was carried out, a Raman study was performed to identify the evolution of the samples in which the surfactant molarity was increased. Figure 7 shows a superposition of the Raman spectra belonging to each sample for the case of one week of storage. It is possible to observe three characteristic peaks: (i) The D band at 1344 cm^−1^ is due to the phonon induced by defects associated with the breakdown of kinematic restriction disorders (breathing mode A1) [46]. This band is attributed to the disorder of the solution because of the presence of vacancies and due to lattice defects caused by the mixture of Sp^2^ and Sp^3^ bonds. (ii) The G band at 1591 cm^−1^ is due to the phonon mode allowed by Raman, which shows the Sp^2^ vibration bonds (carbon–carbon), due to the graphene-type bonds [47]. Finally (iii) the G’ band at 1622 cm^−1^ is related to the second-order dispersion process that can involve two phonons of the same mode (overtone) or phonons of different modes (combinations) having a similar origin of the D band [48]. Using the I_D_/I_G_ relationship, the structural order of the MWCNTs was estimated; the D band intensity decreases with the decrease in the defect density. To calculate the I_D_/I_G_ ratio, the spectra were deconvoluted with Lorentzian functions.

Figure 8 shows the evolution of the I_D_/I_G_ relationship as the molarity increases, showing a tendency to grow. That means that the disorder may increase in the system. According to the literature, spectral properties vary depending on the mechanical conditions of tension by stretching or compression of the MWCNTs and the temperature to which they are subjected. This phenomenon is especially relevant in the case of CNTs mixed with other substances. In this case of the D and G bands in the Raman spectrum of the mixed material, these bands become a sensor of high sensitivity at the microscopic level, giving information about the stress conditions due to the stretching or compression to which they are subjected once they are dispersed [49,50]. Then, as the molarity of the dispersant increases, the stress can increase because of the greater quantity of mass producing greater friction between the nanotubes and dispersant, increasing the disorder. This behavior can be explained by the effect of the dispersant that consists in retaining the separation between MWCNTs when they exhibit a Brownian movement, due to the influence of the ultrasonic tip. For this reason, local cuts are produced in the unraveled MWCNTs, increasing the disorder and then the I_D_/I_G_ relationship; on the other hand, an interaction between surfactant molecules and MWCNTs is produced that increases the friction as the molarity increases [51,52].

This increase in disorder indicates a transformation from Sp^2^ and Sp^3^ bonds in the solution. Then, it can be concluded that the mixtures of MWCNTs, water, and TXT-100 exhibit a lower disorder for lower values of the surfactant molarities.

In previous works, it has been reported that 10 mM is the most suitable molarity [11,53]. It was possible to see that the tense-active molecules act as (i) an exfoliant in the MWCNT agglomerations and (ii) a delay factor in the reagglomeration of MWCNTs in the period of storage. Based on previous works [52] and the stability observed using Z-potential and Raman analyses, three samples of paste cement were prepared, including MWCNTs, 10 mM molarity (TX-100), and 390 J/g sonication energy. Table 2 includes the nomenclature of the test cylinder (samples). This procedure was carried out to evaluate if the mechanical properties are maintained over time. For this study, the samples were cured for 7, 14, and 21 days according to the ASTM-C39 standard.

### Mechanical Test

Stress vs strain curves for each sample were obtained after both one and four weeks of storage. Figure 9 presents a compressive test for S2 and S3, from which the elastic modulus and maximum strength are obtained, according to the procedure described in Figure 10. These results are presented in Figure 11. It can be observed that the Young’s modulus (Figure 11a) and the maximum stress (Figure 11b) increase for the case of samples S2 and S3 compared to the S1 sample.

An experiment was carried out that consists of measuring the elastic modulus and the maximum stress of the material with three different metal components. For this, measurements were made on three different days, i.e., on day 1, a week later, and 15 days later. Therefore, it is valid to use the design of experiments with the factor (S1, S2, S3) and the factor (day 7, 14, 28).

The factorial design of the experiments was used to check the influence of the material and the day of measurement on the properties of the elastic modulus and maximum strength in two separate experiments. For this design of experiments, it is necessary to check the assumptions of normality, homoscedasticity, and randomness. For this, graphical and statistical diagnostic tests were used, which are not presented in this article because they do not expand on the results. Using a significance of σ = 0.05 and, therefore, a confidence of 95, the assumptions of normality and equality of variances were confirmed, as shown in Table 3.

Given that the assumptions for both experiments were met, two ANOVA tables were made to verify whether the day of measurement or the material significantly influence the elasticity or the maximum force. Using a significance of 0.05, that is, a confidence of 95%, the *p*-values of the ANOVA tests carried out to validate the influence of the day and the material on both properties are summarized. It is observed that all the *p*-values are less than 0.05; therefore, it is concluded that both the material and the day of measurement significantly influence the elasticity and maximum strength of the metal.

Using Dunnett’s method with a significance of 0.05, it was observed that in the first measurement, on day 1, significantly lower average values of elasticity and maximum strength were obtained. In addition, it was also verified that the maximum force with material S3 is considerably higher than with materials S1 and S2; however, these two are comparably similar. However, for elasticity, the value for material S1 is less than those for S2 and S3, but these two are considered equivalent.

These results can be explained due to the fact that: (i) The MWCNTs unravel during the sonication process (Figure 10). The MWCNTs (reinforcement phase) exhibit a good interaction with the matrix phase (cement paste) through the radicals available on the surface, establishing good binding within the sample. (ii) There is a shortage of secondary bonds and van der Waals forces (as observed in the UV-Vis analysis); that avoids the presence of energetically weak bonds. By including a surfactant, a more significant amount of carboxylic residues are formed, transforming sp^2^ bonds into sp^3^ (as observed in the Raman analysis) [54]. These bonds interact within the cement matrix with C-S-H phases, generating different bridges (bridge effects) to link capillary pores and inhibit the crack propagation [44,45,55,56]. On the other hand, as the curing time increases, the mechanical properties also increase. This is due to the fact that the hydration is accelerated by the addition of MWCNTs because these act as crystallization centers for the hydrated cement. Moreover, it fills the holes between cement grains, giving as a result an immobilization of water and generating a decrease in the porosity of the samples. Figure 9b shows the behavior of the maximum stress vs. the curing time of the specimens, where the increase in the S2 sample is evident. It manages to reach the maximum effort, so it will have a greater resistance before breaking or reforming. This maximum value of this sample is possible due to it exhibiting a greater stability according to the Z-potential analysis.

In the literature, there are several works that report the enhancement of the mechanical properties of Portland cement by adding MWCNTs; for instance, Yousefi et al. [57] present results showing that the addition of a surfactant employing the mild ultrasonication technique facilitates the homogeneous dispersion of MWCNTs in the cement matrix and enhances the mechanical properties of the hardened concrete. A more recent work by Shahzad et al. [58] reported a study focused on different techniques for dispersing MWCNTs in cementitious materials and the impact on the mechanical properties. As MWCNTs are better dispersed, they tend to fill the micropores, thus increasing the density of the matrix and improving the mechanical properties. Then, it is concluded that there is an enhancement of mechanical properties of MWCNTs’ cement pastes for low sonication energies (less than 1000 J/mL). For higher sonication energies, the mechanical properties decrease, especially because of the higher cohesion of the pastes and the consequent higher difficulty of molding, the incorporation of empty spaces, and/or the higher damages suffered by the MWCNTs [59].

Figure 12 shows an SEM image for sample S3 after mechanical failure. In this image, three forms of arrangement of the MWCNTs can be observed, i.e., bridge effect, spiderweb, and cement fragment without MWCNT anchorage. This image was taken with a scanning electron microscope at resolutions of 2 and 10 µm. Based on the results obtained in the mechanical evaluation, it can be deduced that the bridging effect occurs because a large number of MWCNTs have sufficient length to join capillary pores and act as crack bridging as a result of the covalent bonds, inhibiting the propagation of cracks. This behavior generates a better load capacity, ductility, and fracture energy of the pastes. On the other hand, the spiderweb effect is due to the fact that there was not a total dispersion of nanotubes or, at the time of constructing the test cylinders, a great homogeneity was not reached. 

As a prospective work, a study of the effect of other different dispersants, possibly more affordable, on the physicochemical, electrical, and mechanical properties of the systems including MWCNTs should be carried out.

Regarding future work, considering that corrosion is a process that affects constructions and buildings, it is necessary to carry out an investigation on the corrosion resistance of steel embedded in cement with the addition of MWCNTs.

A great challenge and limitation for these investigations are the MWCNTs, as stated by another author [60]. Despite being used in various areas, carbon nanotubes still have a high cost, which can be an obstacle to the use of this material in cementitious compounds. It is believed that with the increase in demand and with the possibility of synthesizing CNTs for the manufacture of various applications, the material will become more accessible. Thus, even though the cost of the material is currently a negative aspect, the tendency is for this drawback to be overcome over time.

Moreover, according to the literature, the addition of MWCNTs improves several properties of the concrete (comprised of water, aggregates, and cement), including the mechanical properties. The results reported by Mohsen et al. [61] indicated that a high CNT content (greater than 15%) would increase the flexural strength of concrete by more than 100%. Furthermore, the results also showed that CNTs would increase the ductility of concrete by about 150%. Adhikary et al. [62] also reported that the utilization of CNTs significantly improves the mechanical performance of lightweight concrete. An almost 41% improvement in the compression of lightweight concrete was observed at 0.6 wt% CNT loading.

## 4. Conclusions

Multiple wall carbon nanotubes (MWCNTs) were mixed with cement paste and Triton TX-100 surfactant while varying the molarities (10 mM, 20 mM, 40 mM, 60 mM, 80 mM, and 100 mM) and the time of storage (1, 2, 4, 10, and 13 weeks). This procedure was implemented to study the influence of the molarity and time of storage on the mechanical properties of the samples.

The UV-Vis results showed that as the molarity is increased, the intensity of the maximum absorbance (at 300 nm) increases. This peak is due to the presence of the interaction between the surfactant and MWCNTs (π-π stacking and van der Waals forces).

On the other hand, an increasing tendency of the Z-potential as the molarity and storage period were increased was observed. The increase in the molarity generates an increase in the tense-active micelles that decreases both the structural damage and the electrostatic charges present in the MWCNT surface.

According to the Raman results, three characteristic peaks were observed: D band at 1344 cm^−1^ attributed to the disorder of the solution, G band at 1591 cm^−1^ due to the phonon mode of the graphene type bonds, and G’ band at 1622 cm^−1^ related to the second-order dispersion process.

The evolution of the I_D_/I_G_ relationship as the molarity increases is an indication of a growth tendency. This indicates that the disorder increases in the system.

Regarding the mechanical properties, it can be observed that the Young’s modulus and the maximum stress increased for samples of cement + H_2_O + TX-100 with MWCNTs. Furthermore, as the curing time increases, the mechanical properties also increase due to the hydration being accelerated by the MWCNTs.

The SEM image for sample S3 after mechanical failure shows the formation of three arrangements of the MWCNTs: bridge effect, spiderweb, and cement fragment without MWCNT anchorage. The bridging effect occurs because a large number of MWCNTs have sufficient length, acting as a crack bridging as a result of the covalent bonds, inhibiting the propagation of cracks. The spiderweb effect is due to the fact that there was not a total dispersion of nanotubes or, at the time of constructing the test cylinders, a great homogeneity was not reached.

For future work, studies with other surfactants and with concrete containing different aggregates are proposed. In addition, we propose to carry out research with carbon nanotubes produced in our research laboratory. We also propose to make a variation of the sonication energy to reach values close to 1000 J/g.

## Figures and Tables

**Figure 1 materials-15-09035-f001:**
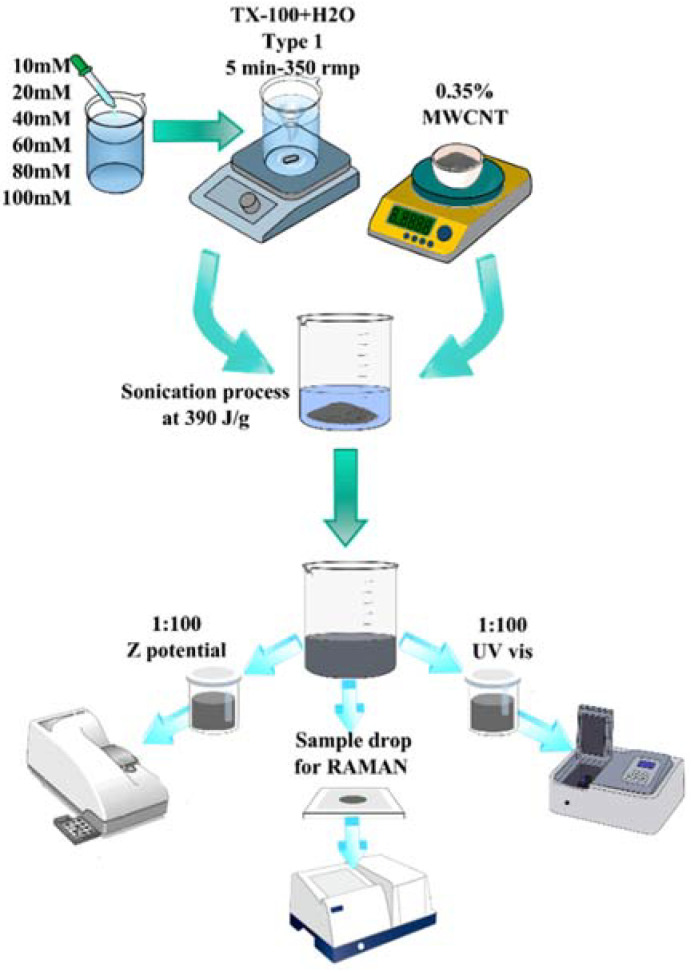
Scheme of the procedure to produce and analyze the dispersion of MWCNTs dissolved in type 1 water, varying the molarity of the surfactant between 10 mM and 100 mM.

**Figure 2 materials-15-09035-f002:**
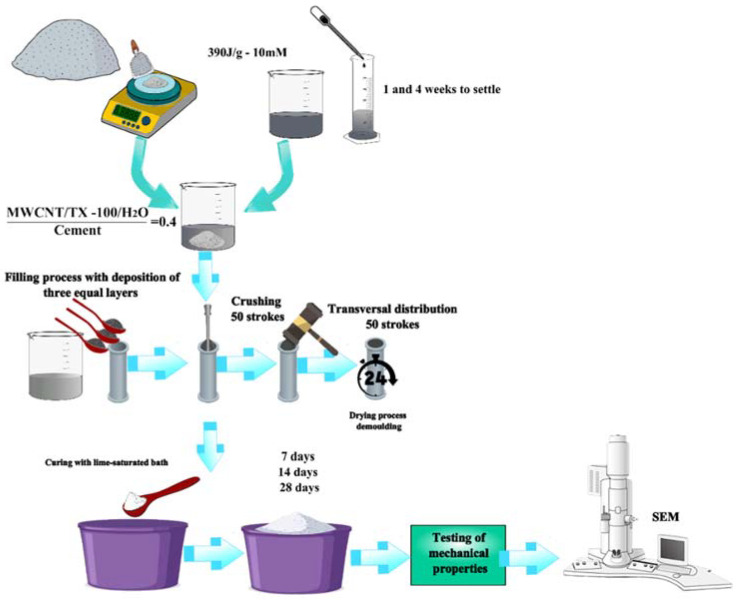
Scheme of the procedure to produce and analyze cement specimens with MWCNT solution dispersed at 390 J/g and with a molarity of surfactant of 10 Mm.

**Figure 3 materials-15-09035-f003:**
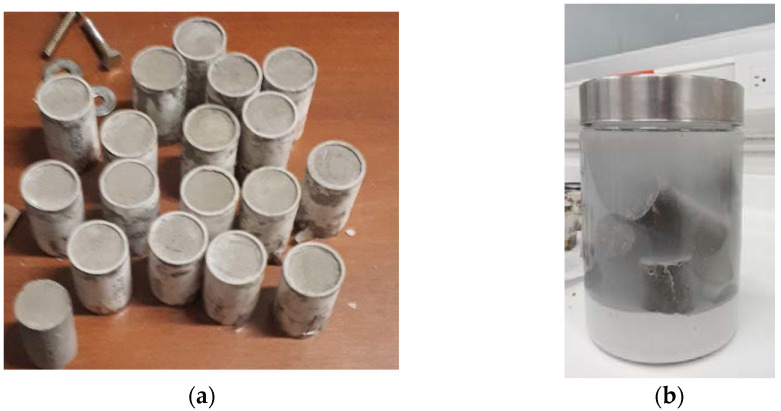
Image of the test cylinder (**a**) during the drying process and (**b**) during the curing and storage process.

**Figure 4 materials-15-09035-f004:**
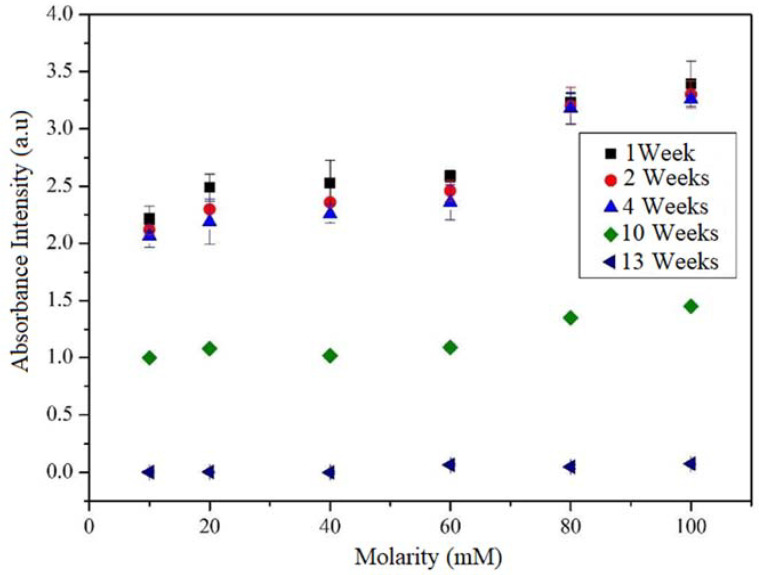
Evolution of the 300 nm absorbance peak depending on the number of weeks of MWCNT dispersion in water with different TX-100 molarities.

**Figure 5 materials-15-09035-f005:**
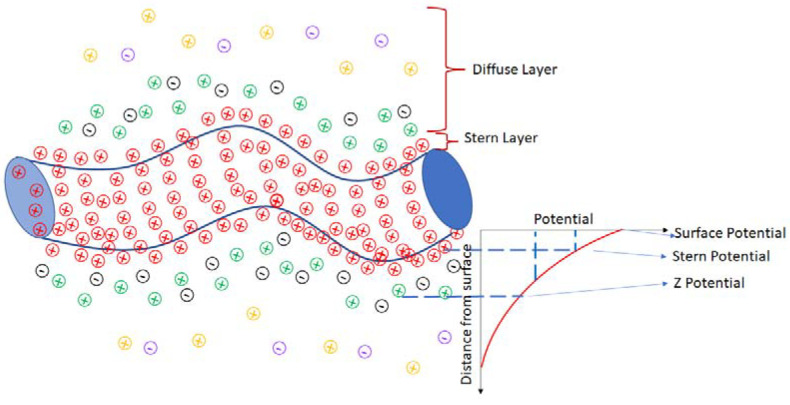
Schematic representation of the zeta potential for nanotubes. The plus and minus signs represent the positive and negative charges, respectively. Colors of charges represent the potentials (charges from surface potential are red; charges from stern potential are green and black; charges from Z-potential are yellow and purple).

**Figure 6 materials-15-09035-f006:**
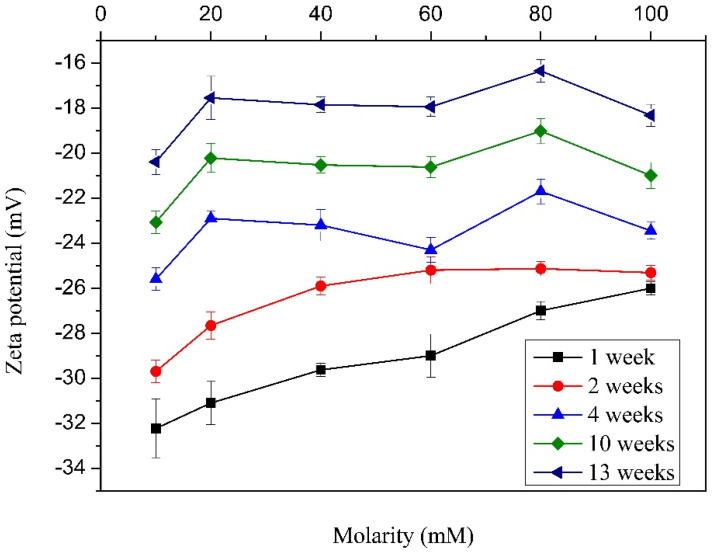
Z-potential of MWCNT dispersions in water with different TX-100 molarities and weeks of storage.

**Figure 7 materials-15-09035-f007:**
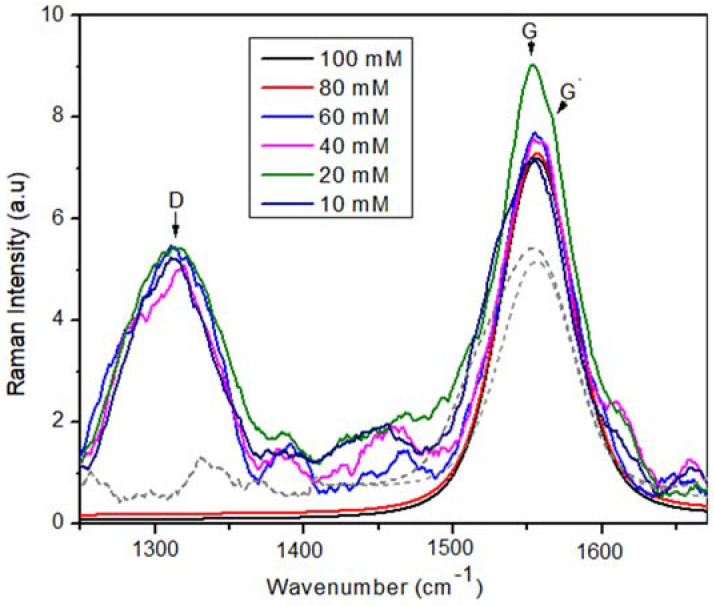
Raman spectra of MWCNT dispersions in water with different TX-100 molarities.

**Figure 8 materials-15-09035-f008:**
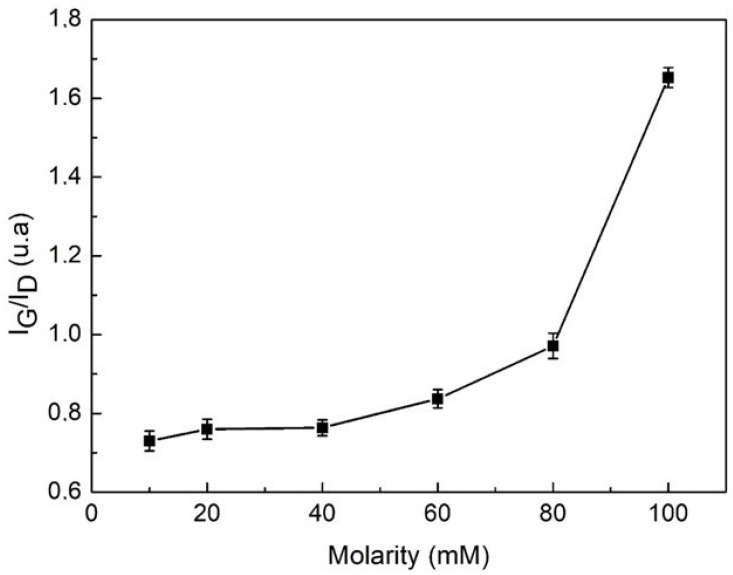
I_D_/I_G_ ratio of MWCNTs dispersed in water with different TX-100 molarities.

**Figure 9 materials-15-09035-f009:**
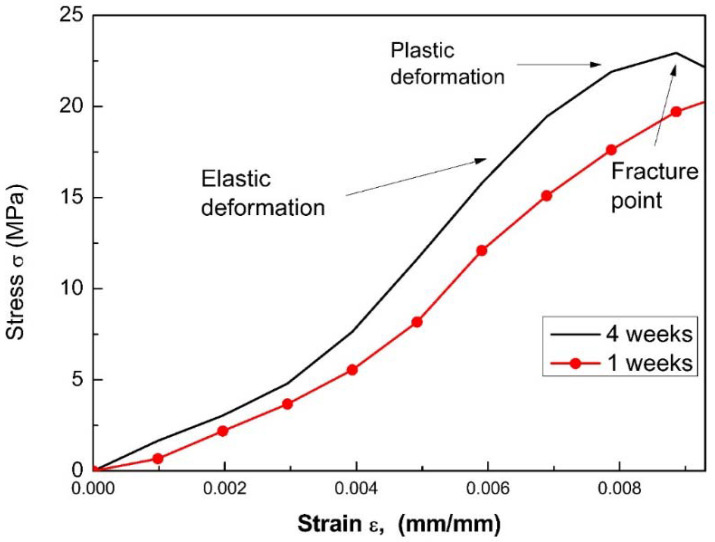
Compressive tests for cement cylinders with the dispersion for S2 and S3.

**Figure 10 materials-15-09035-f010:**
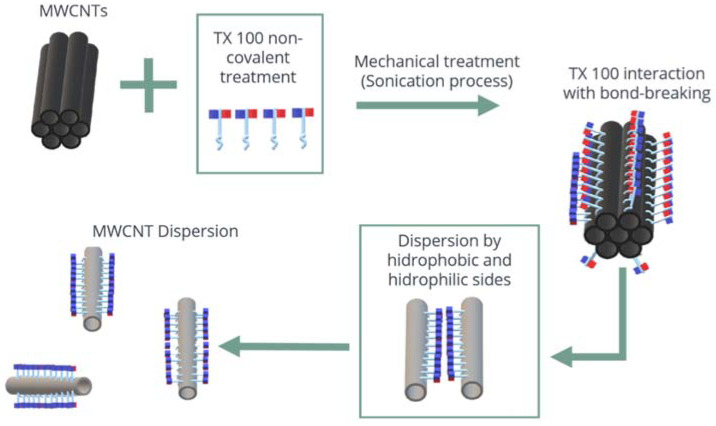
Scheme of the gradual exfoliation processes of the MWCNTs.

**Figure 11 materials-15-09035-f011:**
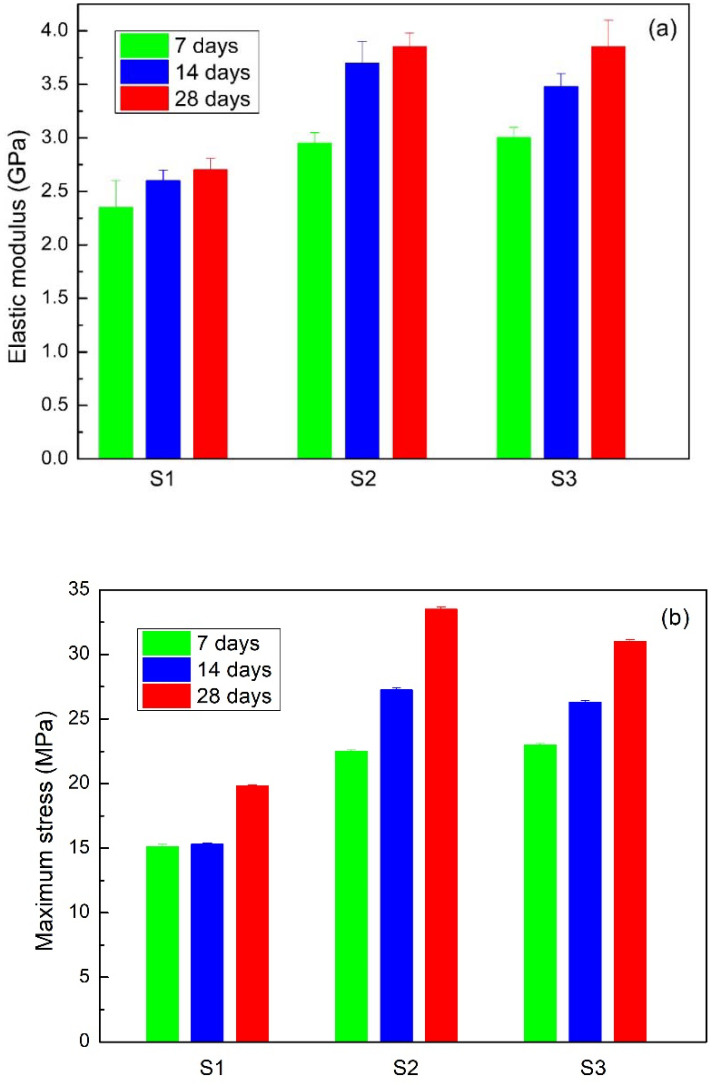
(**a**) Elastic modulus and (**b**) maximum stress to S1, S2, and S3 samples.

**Figure 12 materials-15-09035-f012:**
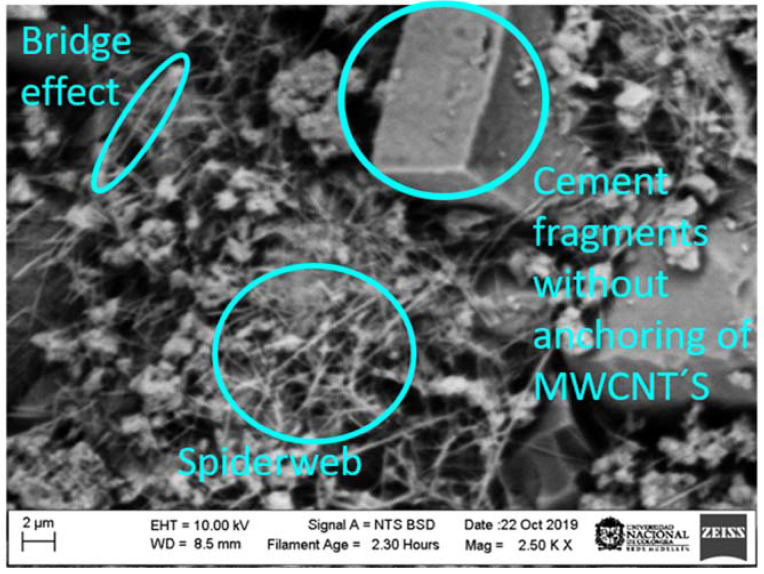
SEM image obtained for a cement paste after the compressive test. This image was taken of sample S3.

**Table 1 materials-15-09035-t001:** Mass of TXT-100, total mass of the mixture, and sonication time for each molarity of TXT-100, following the parameters: energy—390 J/g and 0.35% mass MWCNT of the sample.

Molarity of TXT-100 (mM)	Mass of TXT-100 (g)	Total Mass (g)	Sonication Time (min:s)
10	0.647 ± 0.001	100.997 ± 0.001	39:10 ± 0:0.01
20	1.29 ± 0.001	101.64 ± 0.001	39:35 ± 0:0.01
40	2.59 ± 0.001	102.94 ± 0.001	39:85 ± 0:0.01
60	3.88 ± 0.001	104.23 ± 0.001	40:35 ± 0:0.01
80	5.18 ± 0.001	105.53 ± 0.001	41:25 ± 0:0.01
100	6.47 ± 0.001	106.82 ± 0.001	41:35 ± 0:0.01

**Table 2 materials-15-09035-t002:** Description and notation of the samples used for the mechanical properties characterization.

Description of the Mixtures Used for Building the Test Cylinders	Nomenclature of the Samples
cement + H_2_O + TX-100	S1
cement + H_2_O + TX-100 + MWCNT (one week of storage)	S2
cement + H_2_O + TX-100 + MWCNT (four weeks of storage)	S3

**Table 3 materials-15-09035-t003:** ANOVA for the results obtained for the Young’s modulus and maximum strength.

Factor	*p*-Value	
	Young’s Modulus	Maximum Strength
Sample	0.00215	0.01721
Time of curing	0.01019	0.00414

## Data Availability

Not applicable.

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
