# Peer review of "Effects of Molarity and Storage Time of MWCNTs on the Properties of Cement Paste"

_materials, 2022, doi:10.3390/ma15249035_

Round 1

Reviewer 1 Report

1.Must improve the titile of paper "Study About MWCNT Dispersion (Molarity and Storage Time) 2 To Be Applied in Cement Paste" change to "Effects of molarity and storage time of MWCNT on the properties of cement paste"

2.Must rewrite "inally, an increase in the elastic modulus and maximum strength were obtained when MWCNTs were included in the cement paste. Moreover, it was demonstrated that, at a storage time of four weeks, the mechanical properties of the mixture were retained."

3.Explain more "During all the experiments, the mix was submerged in a cold bath to avoid the sample overheating. The sonication time was calculated with the relationship obtained by Mendoza-Reales (155 gr/60 min) [11]. A diagram of the experimental setup is shown in Figure 1."

4.Please recheck " It is a possible to note that the solution (H2O+TX-100+MWNTC) was stable for four weeks, because, at the 10th and 13th weeks, the intensity of absorbance peak abruptly decreases, reaching a value of 0. 145 This is a promising result because as far as we know, the time for which the solution (H2O+TX-100+MWNT) remains active has not been reported before"

5.Redraw with the novelty of research "At a dispersion of 0.35% MWCNTs into paste cement, it was found that the 10 mM molarity was the most suitable between those used in this work. On the other hand, when paste cement cylinders were built with dispersions stored for four weeks, their mechanical properties were sustained, compared with samples stored for one week. Besides, it was observed that the inclusion of MWCNTs increased the elastic modulus (36.97%) and exhibited a maximum stress by 69.8%."

Author Response

  1. Must improve the title of paper "Study About MWCNT Dispersion (Molarity and Storage Time) 2 To Be Applied in Cement Paste" change to "Effects of molarity and storage time of MWCNT on the properties of cement paste"

Answer: Thanks for your suggestion. Title was changed, and the final title was: “Effects of molarity and storage time of MWCNT on the proper-ties of cement paste”

  1. Must rewrite "finally, an increase in the elastic modulus and maximum strength were obtained when MWCNTs were included in the cement paste. Moreover, it was demonstrated that, at a storage time of four weeks, the mechanical properties of the mixture were retained."

Answer: This expression was rewritten as follows (abstract) “Regarding the mechanical properties, it was observed an increase in the elastic modulus when MWCNTs were included in the cement paste, for all the storage times, exhibiting an increase in elastic modulus and the maximum stress, as a function of the storage time was increased.”

3.Explain more "During all the experiments, the mix was submerged in a cold bath to avoid the sample overheating. The sonication time was calculated with the relationship obtained by Mendoza-Reales (155 gr/60 min) [11]. A diagram of the experimental setup is shown in Figure 1."

Answer: The expression was rewritten and better explained, furthermore, an expression for the sonification time was included in the document; furthermore, a table with the values of masses and sonication time was also included in the document.

4.Please recheck " It is a possible to note that the solution (H2O+TX-100+MWCNT) was stable for four weeks, because, at the 10th and 13th weeks, the intensity of absorbance peak abruptly decreases, reaching a value of 0.145 This is a promising result because as far as we know, the time for which the solution (H2O+TX-100+MWNT) remains active has not been reported before".

Answer: The expression was rewritten; further explanation about the behavior of the UV-Vis spectra and the reasons of the changes in the stability of the mixture was done. The information was included in the document.

  1. Redraw with the novelty of research "At a dispersion of 0.35% MWCNTs into paste cement, it was found that the 10 mM molarity was the most suitable between those used in this work. On the other hand, when paste cement cylinders were built with dispersions stored for four weeks, their mechanical properties were sustained, compared with samples stored for one week. Besides, it was observed that the inclusion of MWCNTs increased the elastic modulus (36.97%) and exhibited a maximum stress by 69.8%."

Answer: This expression was redrawn, improving the conclusions, as in including in the document.

Reviewer 2 Report

This paper investigates the influece of molarity of dispersed MWCNT on the performance of the cementious paste. The research topic should be interesting. And a series of experiments were conducted. However, the results were not well presented, and the analysis and discussion were rather superficial, which reduces the quality of the global manuscript seriously. The reviewer thinks it can not reach the quality of this journal. Some comments are presented as follow:

(1) The expression was over tedious for the language expression.

(2) The experimental preparation should be described in a more detailed way.

(3) The results should be discussed in a more reasonable way.

(4) There was nothing interesting in the conclusion.

Author Response

This paper investigates the influence of molarity of dispersed MWCNT on the performance of the cementious paste. The research topic should be interesting. And a series of experiments were conducted. However, the results were not well presented, and the analysis and discussion were rather superficial, which reduces the quality of the global manuscript seriously. The reviewer thinks it can not reach the quality of this journal. Some comments are presented as follow:

  • The expression was over tedious for the language expression.

Answer: Al the document was revised y rewritten. Professional English correction was done

  • The experimental preparation should be described in a more detailed way.

Answer: More details were added to the experimental preparation. Even, figures of the curing process were included

  • The results should be discussed in a more reasonable way.

Answer: Results were discussed in a more detailed way

  • There was nothing interesting in the conclusion.

Answer: The conclusions were redrawn

Reviewer 3 Report

Study About MWCNT Dispersion (Molarity and Storage Time) To Be Applied in Cement Paste

by

Echeverry-Cardona Laura, Restrepo-Parra Elisabeth, Cabanzo Rafael, and Quintero-Orozco Jorge

General Comment

The manuscript deals with the dispersion of Multiple Walls Carbon Nanotubes (MWCNTs) into cement matrix. In particular, the authors used a Triton TX-100 surfactant with molarities of 10 mM, 20 mM, …, with 0.35% of MWCNTs dispersed in water. I recommend this manuscript for publication in the journal, but in my opinion the paper requires more thorough conclusions.

Specific comments

The following sentences should be revised:

Line 1: Only Article

Line 6: Delete numbers in the keywords

Line 35: add hybridization after sp2

Line 101: g instead of gr

Line 194: add a full stop at the end of the sentence

Line 212-215: Check typos (space after molarity; enter 2 as a subscript)

Line 221: Figure 9 is indicated before Figure 8

Lines 233-234: C-S-H instead of S-C-H

Line 241: Delete cement

Line 242: add a full stop at the end of the sentence

Line 244: add a full stop at the end of the sentence

Line 253: add a full stop at the end of the sentence

Author Response

Study About MWCNT Dispersion (Molarity and Storage Time) To Be Applied in Cement Paste

By: Echeverry-Cardona Laura, Restrepo-Parra Elisabeth, Cabanzo Rafael, and Quintero-Orozco Jorge

General Comment

The manuscript deals with the dispersion of Multiple Walls Carbon Nanotubes (MWCNTs) into cement matrix. In particular, the authors used a Triton TX-100 surfactant with molarities of 10 mM, 20 mM, …, with 0.35% of MWCNTs dispersed in water. I recommend this manuscript for publication in the journal, but in my opinion the paper requires more thorough conclusions.

Specific comments

The following sentences should be revised:

Line 1: Only Article

Answer: Correction done

Line 6: Delete numbers in the keywords

Answer: Correction done

Line 35: add hybridization after sp2

Answer: Correction done

Line 101: g instead of gr

Answer: Correction done

Line 194: add a full stop at the end of the sentence

Answer: Correction done

Line 212-215: Check typos (space after molarity; enter 2 as a subscript)

Answer: Correction done

Line 221: Figure 9 is indicated before Figure 8

Answer: This correction was done

Lines 233-234: C-S-H instead of S-C-H

Answer: Correction done

Line 241: Delete cement

Answer: Correction done

Line 242: add a full stop at the end of the sentence

Answer: Correction done

Line 244: add a full stop at the end of the sentence

Answer: Correction done

Line 253: add a full stop at the end of the sentence

Answer: Correction done

Reviewer 4 Report

The paper has an adequate scope, considering the aims and scope of the Materials Journal. The authors present an interesting study on carbon nano tubes Dispersion (agent dosage amount and storage time) in Cement based materials. In the reviewer's opinion, this is a relevant work, which provides interesting findings, that deserves to be shared with the scientific community. However, and in contrast to the extensive scientific efforts, the presentation of the manuscript does not comply with the standards of a publication such as Materials. Therefore, the reviewer suggests the authors to prepare a corrected version by carrying out an extensive edition based on the recommendations provided below. The following suggestions and comments should be taken into account before accepting the article for publication:

1. Overall, the experimental methods used are well known and the results are not explained in detail. It seems many experiments have been carried out and their results are summarized without in-depth discussion and comparison with other studies on the same subject area. 

2. Please show the scatters all presented experimental results. This applies to Figures (via error bars) and also in the related Tables (by adding the +/- variabilities).

3. Results should include the measurement uncertainty, and expressed as mean value +/- (1x or 2x) standard deviation.

4. Please discuss the differences between the results considering the statistical significance (e.g. using ANOVA, t-test). For example this is in particular needed to discuss the changes in main results and state if the conclusions are statisticaly significant. Discussion of results and Conclusions should be thus revised.

5. Another critical concern is that the authors have merely reported the observations. Further discussion on the result be included and in-depth analysis be made. Prospects, challenges, future work, limitations, etc. must be discussed in this section.

6. The work is focused on cement pastes only; however cementitious materials almost always include aggregates in practical applications. What would be the effect of aggregates? Please discuss this. 

7. Please also add future research steps which will follow this work.

In the Reviewer's opinion the manuscript should be published in the journal after major revision.

Author Response

The paper has an adequate scope, considering the aims and scope of the Materials Journal. The authors present an interesting study on carbon nano tubes Dispersion (agent dosage amount and storage time) in Cement based materials. In the reviewer's opinion, this is a relevant work, which provides interesting findings, that deserves to be shared with the scientific community. However, and in contrast to the extensive scientific efforts, the presentation of the manuscript does not comply with the standards of a publication such as Materials. Therefore, the reviewer suggests the authors to prepare a corrected version by carrying out an extensive edition based on the recommendations provided below. The following suggestions and comments should be taken into account before accepting the article for publication:

  1. Overall, the experimental methods used are well known and the results are not explained in detail. It seems many experiments have been carried out and their results are summarized without in-depth discussion and comparison with other studies on the same subject area. 

Answer: The experimental details were organized, and results were better discussed, and compared with other works. This comparison was done at the end of the section of mechanical properties.

  1. Please show the scatters all presented experimental results. This applies to Figures (via error bars) and also in the related Tables (by adding the +/- variabilities).

Answer: In tables, the +/- variabilities were included, and all figures included error bars, except figure 8, since the error is in the size of the mark.

  1. Results should include the measurement uncertainty, and expressed as mean value +/- (1x or 2x) standard deviation.

Answer: The deviation was included in tables and figures; except results of figure 8, where the deviation is in the order or the mark

  1. Please discuss the differences between the results considering the statistical significance (e.g. using ANOVA, t-test). For example, this is in particular needed to discuss the changes in main results and state if the conclusions are statistically significant. Discussion of results and Conclusions should be thus revised.

Answer: Thank you for the correction. ANOVA was included for the results of mechanical properties

  1. Another critical concern is that the authors have merely reported the observations. Further discussion on the result be included and in-depth analysis be made. Prospects, challenges, future work, limitations, etc. must be discussed in this section.

Answer: Tank you to the referee. The next text was included: As a prospective work, a study of the effect of other different dispersants, possibly more affordable, on the physicochemical, electrical and mechanical properties of the systems including MWCNTs should be carried out.

Regarding future work, considering that corrosion is a process that affects constructions and buildings, it is necessary to carry out an investigation on the corrosion resistance of steel embedded in cement with the addition of MWCNTs.

A great challenge and limitation for these investigations is the MWCNTs. Despite being used in various areas, carbon nanotubes still have a high cost, which can be an obstacle to the use of this material in cementitious compounds. It is believed that with the increase in demand and with the possibility of synthesizing NTCs for the manufacture of various applications, the material will become more accessible. Thus, even though the cost of the material is currently a negative aspect, the tendency is for this drawback to be overcome over time.

  1. The work is focused on cement pastes only; however cementitious materials almost always include aggregates in practical applications. What would be the effect of aggregates? Please discuss this. 

Answer: Several authors have reported that addition of CNTs improves the mechanical and other properties behavior of concrete (comprises of water, aggregates, and cement); some of these reports were included at the end of the results and analyses

  1. Please also add future research steps which will follow this work.

Answer: Future works were included in the conclusions 

Round 2

Reviewer 1 Report

It can be published. 

Reviewer 4 Report

Authors have well addressed most of the raised issues and sufficiently improved the manuscript. Now, I recommend this work for publication.

Congrats to the authors. Best whishes on the research impact.